

# The *Plasmodium berghei* RC strain is highly diverged and harbors putatively novel drug resistance variants

Warangkhana Songsungthong[1,2], Supasak Kulawonganunchai[3], Alisa Wilantho[3], Sissades Tongsima[3], Pongpisid Koonyosying[1], Chairat Uthaipibull[1], Sumalee Kamchonwongpaisan[1] and Philip J. Shaw[1]

[1] Protein-Ligand Engineering and Molecular Biology Laboratory, Medical Molecular Biology Research Unit, National Center for Genetic Engineering and Biotechnology (BIOTEC), National Science and Technology Development Agency (NSTDA), Pathum Thani, Thailand
[2] Biomolecular Analysis and Application Laboratory, Biosensing Technology Research Unit, National Center for Genetic Engineering and Biotechnology (BIOTEC), National Science and Technology Development Agency (NSTDA), Pathum Thani, Thailand
[3] Biostatistics and Bioinformatics Laboratory, Genome Technology Research Unit, National Center for Genetic Engineering and Biotechnology (BIOTEC), National Science and Technology Development Agency (NSTDA), Pathum Thani, Thailand

## ABSTRACT

**Background:** The current first line drugs for treating uncomplicated malaria are artemisinin (ART) combination therapies. However, *Plasmodium falciparum* parasites resistant to ART and partner drugs are spreading, which threatens malaria control efforts. Rodent malaria species are useful models for understanding antimalarial resistance, in particular genetic variants responsible for cross resistance to different compounds.

**Methods:** The *Plasmodium berghei* RC strain (*Pb*RC) is described as resistant to different antimalarials, including chloroquine (CQ) and ART. In an attempt to identify the genetic basis for the antimalarial resistance trait in *Pb*RC, its genome was sequenced and compared with five other previously sequenced *P. berghei* strains.

**Results:** We found that *Pb*RC is eight-fold less sensitive to the ART derivative artesunate than the reference strain *Pb*ANKA. The genome of *Pb*RC is markedly different from other strains, and 6,974 single nucleotide variants private to *Pb*RC were identified. Among these *Pb*RC private variants, non-synonymous changes were identified in genes known to modulate antimalarial sensitivity in rodent malaria species, including notably the ubiquitin carboxyl-terminal hydrolase 1 gene. However, no variants were found in some genes with strong evidence of association with ART resistance in *P. falciparum* such as K13 propeller protein.

**Discussion:** The variants identified in *Pb*RC provide insight into *P. berghei* genome diversity and genetic factors that could modulate CQ and ART resistance in *Plasmodium* spp.

Corresponding authors
Sumalee Kamchonwongpaisan, sumaleek@biotec.or.th
Philip J. Shaw, philip@biotec.or.th

## INTRODUCTION

The incidence of malaria is declining around the world, and efforts are being directed toward elimination of this disease in many endemic areas (*Tanner et al., 2015*; *WHO Malaria Policy Advisory Committee and Secretariat, 2015*). However, malaria parasite resistance to first-line artemisinin (ART) combination therapy is evolving in Southeast Asia (*Woodrow & White, 2016*), including resistance to partner drugs such as piperaquine (*Duru, Witkowski & Ménard, 2016*; *Amato et al., 2016*; *Imwong et al., 2017*). This is alarming given that parasites resistant to chloroquine (CQ) (*Wootton et al., 2002*) and pyrimethamine (*Nair et al., 2003*) are highly prevalent in this region. The specter of multi-drug resistant *Plasmodium falciparum* parasites could undermine all recent advances in reducing the disease burden. Laboratory models of antimalarial resistance are needed to develop new drugs effective against parasites resistant to currently available antimalarials, and to understand the molecular mechanisms of resistance.

Rodent malaria parasites are widely used laboratory models for human malaria as they can be studied in vivo in animal and mosquito hosts. Antimalarial-resistant parasites can be selected by repeated dosing of infected animals, and stably resistant parasite clones can be isolated after serial passage in animal hosts. Moreover, parasites cross-resistant to different drugs have been obtained by this approach. CQ and ART resistant *Plasmodium chabaudi* rodent malaria parasites from laboratory selection were isolated in *Afonso et al. (2006)*. Laboratory selected CQ and ART-resistant *Plasmodium yoelii* rodent malaria parasites display impaired hemozoin production and elevated level of glutathione (GSH), suggestive of a common mechanism of resistance against the two drugs (*Witkowski et al., 2012*). Laboratory selection of antimalarial resistance has also been performed for the most virulent rodent malaria species *Plasmodium berghei*. The *P. berghei* RC strain (*Pb*RC) was obtained by laboratory selection with CQ, and is defective for production of hemozoin (*Peters, 1964*; *Peters, Fletcher & Staeubli, 1965*). This strain is also reported as resistant to other drugs, including ART (*Pérez-Rosado et al., 2002*). Given that *Pb*RC also shows an elevated level of GSH (*Vega-Rodríguez et al., 2015*), similar to cross-resistant *P. yoelii* (*Witkowski et al., 2012*), it may harbor resistance mutations in the same genes as other drug-resistant rodent malaria parasites.

In this study, we found that *Pb*RC is resistant to artesunate, a water-soluble derivative of ART. We sought the genetic factors responsible for the cross-drug resistant phenotype of *Pb*RC by performing whole genome sequencing. The *Pb*RC genome is markedly different from other characterized *P. berghei* strains, and we identified several variants unique to *Pb*RC in genes which may modulate antimalarial resistance.

## MATERIALS AND METHODS

Female BALB/c mice of about 6 to 10 weeks old were purchased from the National Laboratory Animal Center, Mahidol University, Thailand. The following parasites were obtained from the Malaria Research and Reference Reagent Resource Center (MR4; http://www.beiresources.org), a part of BEI Resources, NIAID, NIH: *Plasmodium berghei* RC, MRA-404, deposited by W. Peters and B.L. Robinson; *Plasmodium berghei* ANKA 507m6cl1, MRA-867 deposited by C.J. Janse and A.P. Waters. Artesunate was a gift

from Dafra Pharma, Turnhout, Belgium. Other reagents, unless otherwise noted, were purchased from Sigma Aldrich (St. Louis, MO, USA).

## Four-day suppressive test for in vivo drug sensitivity

BALB/c mice were injected intravenously with $1 \times 10^7$ *P. berghei* infected red blood cells. For *Pb*RC, five to six mice were used per group. For *Pb*ANKA, four to eight mice were used per group. Oral doses of artesunate were given at 4, 24, 48 and 72 h post infection. Four days post infection, parasitemia was determined by manually counting infected and uninfected red blood cells in Giemsa-stained thin blood smears. Percent inhibition was calculated using the following formula:

$$\text{Percent inhibition} = 100 - ((100 \times \text{parasitemia of each dose})/ \text{parasitemia of untreated control})$$

Data of percent inhibition at different doses of artesunate were fitted to the two-parameter sigmoidal dose response equation using the drc package in R (*Ritz & Streibig, 2005*). This study was carried out in accordance with the guidelines in the Guide for the Care and Use of Laboratory Animals of the National Research Council, Thailand. All animal experiments were performed with the approval of BIOTEC's Institutional Animal Care and Use Committee (Permit number BT-Animal 02/2557). At the end of the experiments, mice were euthanized by $CO_2$ asphyxiation. All efforts were made to alleviate pain and suffering.

## Sanger dideoxy sequencing of PCR products from selected regions

Selected genomic regions with candidate variants in the *P. berghei* RC strain were PCR-amplified using high-fidelity Phusion DNA polymerase (New England Biolabs, Ipswich, MA, USA) using primers listed in Table S1. The resulting PCR products were sent for Sanger dideoxy sequencing (1st BASE, Selangor, Malaysia).

## Whole genome DNA sequencing

Parasitized blood obtained from a single mouse infected with the *Pb*RC parasite was passed through a CF11 (Whatman/Sigma Aldrich, St. Louis, MO, USA) column to remove white blood cells. Cells were harvested by centrifugation and parasites were liberated from red cells by lysis of the red blood cell membrane with 0.2% saponin. Parasites were washed twice with phosphate buffered saline (137 mM NaCl, 2.7 mM KCl, 4.3 mM $Na_2HPO_4$, 1.47 mM $KH_2PO_4$, pH 7.4), resuspended in lysis buffer (8.5 mM Tris–HCl, 342 mM NaCl, 2 mM $Na_2$-EDTA, 0.14 mg/ml proteinase K, 0.14% SDS, pH 8.2) and incubated at 37 °C overnight. Proteins were precipitated by the addition of NaCl to 1.5 M, centrifuged and discarded. Isopropanol was added to the supernatant to precipitate nucleic acids. The nucleic acid pellet was dried and resuspended in TE buffer (10 mM Tris, 1 mM EDTA, pH 8.0). RNA was removed by digestion with RNAseA. Phenol chloroform extraction and ethanol precipitation of DNA were performed. The integrity of genomic DNA was checked by agarose gel electrophoresis. Genomic DNA was submitted to the Chulalongkorn Medical Research Center, Bangkok, Thailand for genome sequencing. Genomic DNA was sheared by

sonication and DNA fragments <1 kb were gel-purified. Sequencing libraries were constructed from sheared genomic DNA using a TruSeq kit (Illumina, San Diego, USA). About $2 \times 150$ bp reads were obtained using a MiSeq instrument (Illumina). The raw data are deposited in the NCBI Sequence Read Archive, accession number PRJNA277169.

## Sequence data analysis

Raw sequencing data were obtained in FASTQ format. Genome sequence data for strains K173, NK65NY, NK65E, SP11_RLL, SP11_A (*Otto et al., 2014*) were downloaded from the European Nucleotide Archive. Filtering of raw data to remove poor quality reads (average Q-score <30) and removal of adapter sequences were performed using FASTQC (*Andrews, 2010*). Preprocessed read data were aligned to the *P. berghei* ANKA version 3 reference genome (downloaded from GeneDB; *Logan-Klumpler et al., 2012*) using BOWTIE version 2.2.2.6 software (*Langmead et al., 2009*) under the setting of length of seed substring 22 bases without clipping. GATK software version 3.3.0 (*McKenna et al., 2010*) was used to identify potential base-substitution single nucleotide variants (SNV) and small insertion/deletions (INDEL) between all strains different from the reference strain by the HaplotypeCaller method combined with the GenotypeGVCFs method, under the settings: –minReadPerAlignmentStart (10), –min_base_quality_score (10), –sample_ploidy(2), –heterozygosity(0.001). SnpEff software version 4.3g (*Cingolani et al., 2012*) was used to annotate the variants using the genome annotation file downloaded from GeneDB. The .vcf file containing all raw variants called by GATK is provided in Data S1. The raw variants were filtered to retain only high-confidence SNVs for genome analyses. INDEL variants were not included, since many occur within simple sequence repeats that could be prone to PCR and sequencing artifacts. Potential false positive SNVs resulting from sequencing artifacts were removed by excluding variants with heterozygous or missing genotypic calls in any strain. Potential false variants resulting from read alignment error in repetitive regions were removed using the DustMasker program (*Morgulis et al., 2006*). Variants present among multigene families listed in Additional File 4 of *Otto et al. (2014)* were also treated as potential false positives and were removed. After filtering, 8,681 SNV markers remained.

Principal components analysis (PCA) was performed using the ipPCA tool (*Limpiti et al., 2011*), implemented in MATLAB version R2009b. The 8,681 filtered SNV markers were used as input for ipPCA. Tajima's D scores were calculated using PopGenome, a population genomic analysis tool in R (*Pfeifer et al., 2014*). Scores were calculated in sliding non-overlapping genomic windows of five consecutive variants (1,730 windows in total), and separately for annotated genes (1,765 genes in total, Table S2). Plots of Tajima's D scores were made using the ggplot2 package in R (*Wickham, 2009*). Gene ontology analysis of genes with Tajima D scores greater than 1 was performed using the Gene ontology web service provided in the PlasmoDB website (*Aurrecoechea et al., 2009*). Terms were considered significant using a Bonferroni-corrected *p*-value threshold of 0.05.

Copy number variants (CNVs) were identified from the whole-genome sequencing data using the cn.MOPS tool implemented in R (*Klambauer et al., 2012*). The sequence data for all six strains aligned to the reference genome were used as input for cn.MOPS,

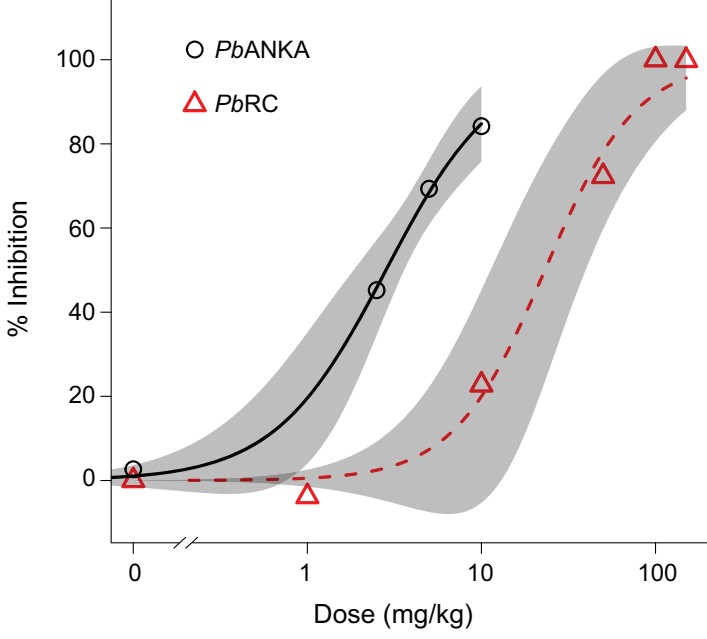

**Figure 1 Artesunate sensitivity of *P. berghei* RC (*PbRC*) in infected mice.** BALB/c mice were infected intravenously with $1 \times 10^7$ infected red blood cells. Artesunate was given orally and parasitemia was determined four days post infection. Five to six mice per group were used for *PbRC* and four to eight mice per group were used for *PbANKA* reference strain. Percent inhibition compared with untreated control (0%) was calculated for each dose of drug. Data were fitted to the two-parameter variable-slope sigmoidal dose–response equation. The average percent inhibitions for each dose are plotted, and the gray bars represent confidence regions calculated by the drc package.

which was run using the haploid genome setting and window size of 500 bp used for local modelling of read counts. Mapped reads in genomic regions with putative variants were visualized using the Integrative Genomics Viewer program (*Thorvaldsdottir, Robinson & Mesirov, 2013*).

# RESULTS

The *PbRC* parasite is reported as CQ and ART resistant (*Pérez-Rosado et al., 2002*). We tested the sensitivity of the *PbRC* strain to artesunate, a water soluble derivative of ART. *PbRC* is approximately eight-fold less sensitive to artesunate compared with *PbANKA* (effective dose for 50% inhibition of parasite ($ED_{50}$) of 2.8 (s.e. = 0.4) and 23.1 (s.e. = 7.3) mg/kg in *PbANKA* and *PbRC*, respectively; Fig. 1), which confirms the multi-drug resistant phenotype for this strain.

## *PbRC* is widely diverged from other *P. berghei* strains

Illumina sequencing of *PbRC* genomic DNA was performed. Sequence reads were obtained from 17,801,263 clusters, and 80.5% of the preprocessed reads could be mapped to the *PbANKA* v3 reference genome. Genomic sequence data of previously sequenced strains K173, NK65NY, NK65E, SP11_RLL and SP11_A (*Otto et al., 2014*) were aligned to the reference genome using the same analytical procedure. SNV and small INDEL variants were called from the mapped reads for all strains. A total of 27,495 variant markers

**Table 1 Summary of *P. berghei* genomic variants identified from whole genome sequencing.**

|  | *Pb*RC | *Pb*K173 | *Pb*NK65_E | *Pb*NK65_NY | *Pb*SP11_A | *Pb*SP11_RLL |
|---|---|---|---|---|---|---|
| Raw[a] | 20,399 | 5,263 | 2,034 | 2,480 | 1,561 | 5,834 |
| Private[b] | 16,774 | 1,991 | 305 | 561 | 292 | 1,693 |
| Missing[c] | 1,295 | 1,326 | 241 | 259 | 224 | 654 |
| Filtered[d] | 7,726 | 1,251 | 12 | 33 | 1 | 890 |
| Private[b] | 6,974 | 504 | 1 | 17 | 1 | 311 |

**Notes:**

The *Pb*RC strain was sequenced in this study; data from other strains were reported in *Otto et al. (2014)*.
[a] Single nucleotide variants (SNV) and small insertion/deletion (INDEL) markers called by the GATK tool using default parameters.
[b] Markers with variant allele detected in only one strain.
[c] Markers with no genotype calls owing to insufficient mapped reads.
[d] SNV remaining after filtering to remove variants with heterozygous calls, located in repetitive regions, or present in multigene families.

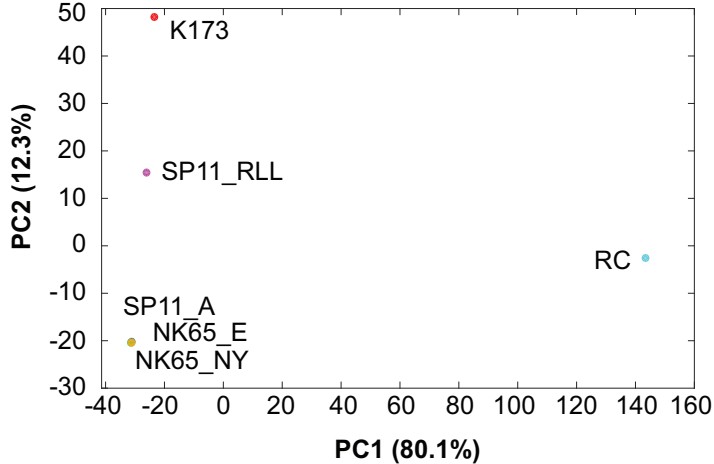

**Figure 2 Principal components analysis of *P. berghei* strains.** The genotypic data from 8,681 single nucleotide variants called from whole genome sequences were encoded as a matrix for principal components analysis. The loadings from the first and second principal components for each strain are plotted.

were identified among the six strains (Data S1), of which 8,681 SNV markers remained after applying stringent filtering criteria. The numbers of variants identified in each strain are shown in Table 1.

The majority of filtered SNVs are private to each strain, and the *Pb*RC strain shows the highest number of variants, indicating that it is markedly different from the other strains. This is confirmed by PCA in which *Pb*RC is clearly separated from the other strains in the first principal component, which captures most of the genotypic variance among these strains (Fig. 2).

## Genetic diversity across the *P. berghei* genome

A previous analysis of *P. berghei* genotypic diversity revealed a low SNV frequency compared with *P. chabaudi*, another rodent malaria species (*Otto et al., 2014*). However, the extreme divergence of *Pb*RC from other strains may provide further insight into *P. berghei* intraspecific genetic diversity. The Tajima's D score was calculated for sliding
genomic windows along all chromosomes, and also for annotated genes. Tajima's D is a measure of the observed versus expected genetic diversity, calculated as the difference between the mean number of pairwise differences and the number segregating sites (*Tajima, 1989*). The mean Tajima's D is negative for each chromosome, indicating a genome-wide pattern of negative Tajima's D (Fig. 3A). However, there are several regions with positive Tajima's D values that are spread throughout the genome (Fig. 3B). At the gene level, 122 genes were identified with a Tajima's D score greater than one (Table S2). No specific gene ontology terms or biological processes are significantly enriched among these genes with high Tajima's D scores. However, a few genes homologous to *Plasmodium* genes with known functions in host cell invasion showed high Tajima's D scores, including PBANKA_132170 (berghepain 1), PBANKA_041290 (circumsporozoite- and TRAP-related protein, *Pb*CTRP), PBANKA_1115300 (glideosome-associated protein 40, *Pb*GAP40) and PBANKA_1137800 (glideosome-associated connector, *Pb*GAC).

Next, variants private to *Pb*RC located within genes were examined to identify putative causal variants of the drug resistant trait in this strain. To our knowledge, the only prior evidence of *Pb*RC genetic variants compared with the *Pb*ANKA reference is an Ile to Lys substitution at residue 413 (I413K) of the gamma glutamylcysteine synthetase (*Pb*γ-*gcs*) gene (*Pérez-Rosado et al., 2002*). We confirmed this mutation in *Pb*RC from whole genome and Sanger sequencing. Comparison of *P. berghei* strain genomes identified this variant as private to *Pb*RC (Table 2). A previous study reported a possible translocation of the multidrug resistance associated protein (MRP) gene to chromosome 8 in *Pb*RC (*Gonzalez-Pons et al., 2009*). However, visualization of *Pb*RC mapped reads showed contiguity in chromosome 14 in the vicinity of the MRP gene reference location, and thus no evidence of translocation (Fig. S1).

We searched for candidate drug resistance variants in genes known to modulate CQ and/or ART drug resistance in other *Plasmodium* spp. The variants in these genes private to *Pb*RC are shown in Table 2. Mutations in the *P. chabaudi* ubiquitin carboxyl-terminal hydrolase 1 (*ubp1*) gene modulate CQ and ART resistance in laboratory-selected drug resistant parasites (*Hunt et al., 2007*; *Henriques et al., 2013*). Five non-synonymous variants private to *Pb*RC were found in the homologous *Pbubp1* gene. Mutations in the CQ resistance transporter (*crt*) gene modulate CQ resistance in *P. falciparum* (*Martin & Kirk, 2004*; *Ecker et al., 2012*). A non-synonymous V42F variant private to *Pb*RC was found in the *P. berghei* homologue (*Pbcrt*). Mutation of the *P. falciparum* multidrug resistance associated protein 1 (*mdr1*) gene modulates sensitivity to several antimalarials, including CQ and ART (*Sanchez et al., 2010*). A non-synonymous V54A variant private to *Pb*RC was found in the homologous *Pbmdr1* gene. Among other proteins implicated as modulators of ART sensitivity in *P. falciparum*, a non-synonymous F320C variant private to *Pb*RC was found in the *Pbpi3k* gene homologous to phosphatidylinositol-3-kinase (*Pf*PI3K) (*Mbengue et al., 2015*). *Pb*RC mutations were notably absent from some genes strongly implicated as modulators of ART sensitivity, including the homologues of μ subunit of adaptor protein 2 complex (AP2-μ; PBANKA_1433900), a gene mutated in the *P. chabaudi* CQ- and ART-resistant AS-ART strain (*Henriques et al., 2013*) and K13 Kelch

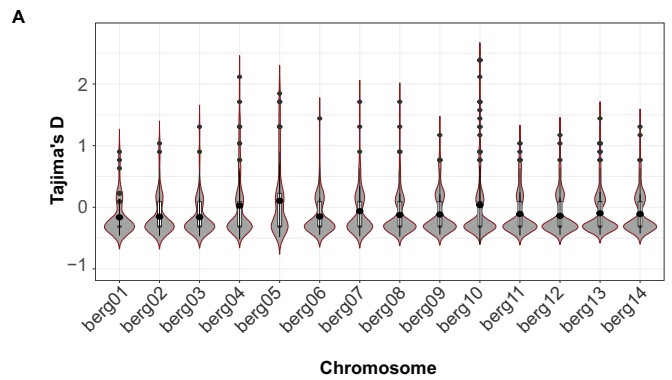

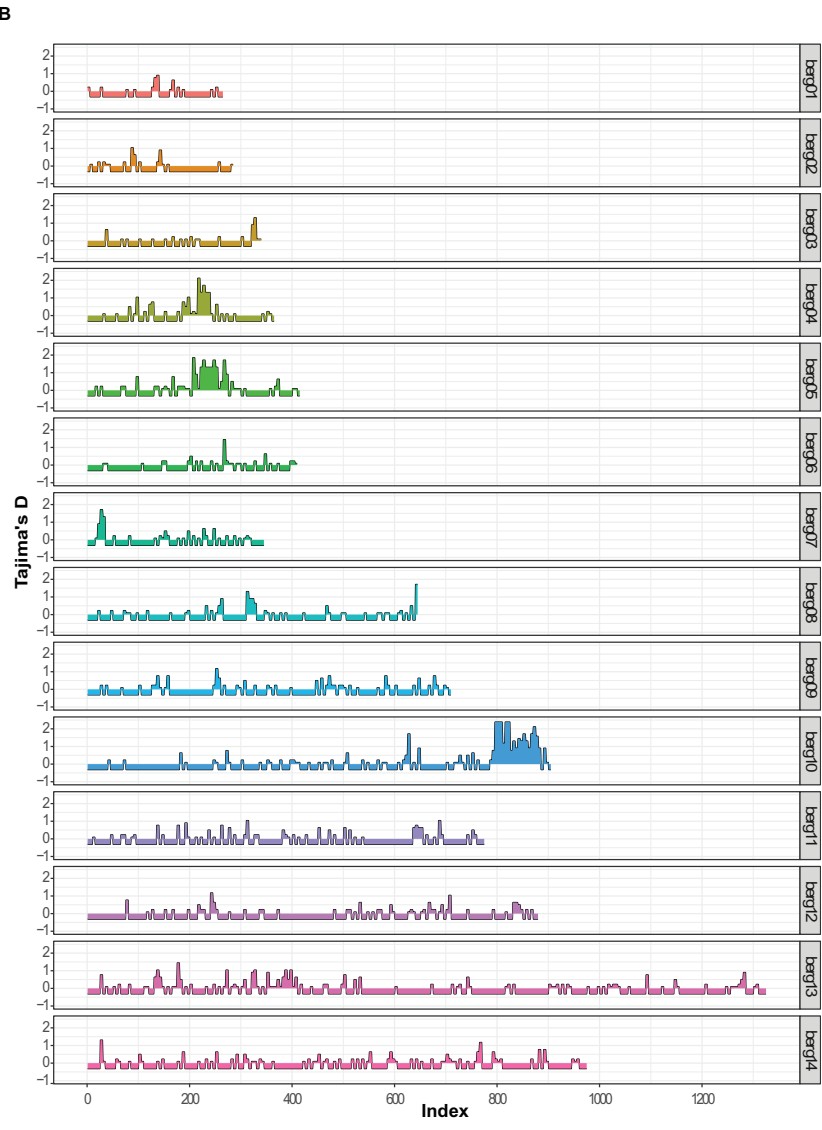

**Figure 3  Genome-wide distribution of Tajima's D scores in *P. berghei*.** Tajima's D score was calculated for sliding, non-overlapping genomic windows of five variants along all 14 chromosomes using the PopGenome tool. Scores were calculated for a total of 1,730 windows from six strains compared with the reference strain *Pb*ANKA v3. The distributions of scores in each chromosome are shown as violin plots in part (A). The scores in each window are plotted separately for each chromosome in part (B).

**Table 2 Non-synonymous variants private to *P. berghei* RC present in genes previously associated with chloroquine and/or artemisinin resistance in *Plasmodium* spp.**

| Gene description | Gene symbol | Gene ID | Missense variants private to *Pb*RC | Confirmed by Sanger dideoxy sequencing |
|---|---|---|---|---|
| γ-Glutamylcysteine synthetase | *Pbγ-gcs* | PBANKA_0819800 | I413K[a] | YES |
| Ubiquitin carboxyl-terminal hydrolase 1 | *Pbubp1* | PBANKA_0208800 | R1561K | YES |
| | | | K1582E[a] | YES |
| | | | K2102E | YES |
| | | | N2279D | YES |
| | | | A2402V | YES |
| Chloroquine resistance transporter | *Pbcrt* | PBANKA_1219500 | V42F | YES |
| Multidrug resistance gene | *Pbmdr1* | PBANKA_1237800 | V54A | YES |
| Phosphatidyl-inositol-3-phosphate kinase | *Pbpi3k* | PBANKA_1114900 | F320C | N.D.[b] |
| Dipeptidyl aminopeptidase 3 | *Pbdpap3* | PBANKA_1002400 | L3I | N.D.[b] |

**Notes:**
Single nucleotide variants (SNV) private to the *P. berghei* RC strain were identified by whole-genome sequencing comparing with the reference strain *Pb*ANKA v3 and other strains reported in *Otto et al. (2014)*.
[a] Raw variants called by GATK, but removed by DustMasker filtering (variant located within a repetitive region).
[b] Sanger dideoxy sequencing not done.

propeller (*Pb*K13; PBANKA_1356700), mutations of which confer reduced ART sensitivity in *P. falciparum* (*Ariey et al., 2013*; *Ghorbal et al., 2014*).

Catabolism of hemoglobin is tied to the mechanisms of action of both CQ and ART. CQ prevents crystallization of heme, a product of hemoglobin catabolism (*Martin & Kirk, 2004*), whereas the antimalarial effect of ART is strongly dependent on catabolism of hemoglobin (*Klonis et al., 2011*; *Xie et al., 2015*). Given that the CQ and ART-resistant *Pb*RC is defective for hemoglobin catabolism and hemozoin formation (*Peters, 1964*; *Peters, Fletcher & Staeubli, 1965*), we investigated whether *Pb*RC harbored private variants in genes encoding proteins known to function in catabolism of hemoglobin (*Ponsuwanna et al., 2016*; *Lin et al., 2015*). Among these genes, a non-synonymous variant private to *Pb*RC was found only in the dipeptidyl aminopeptidase 3 gene (*Pbdpap3*) (Table 2).

In addition to genes previously associated with CQ and ART-resistance, *Pb*RC private variants in other genes may be responsible for the traits associated with this strain. *Pb*RC does not produce gametocytes (*Peters, Fletcher & Staeubli, 1965*), and so may harbor variants in genes important for gametocytogenesis. *Pb*RC is also reported as slow growing compared with the drug-sensitive N strain (*Peters, 1964*), and so may harbor variants in genes important for growth. Furthermore, the ART resistance trait of *Pb*RC may involve mutations in several genes, since ART resistance in *P. falciparum* is a heritable trait in which the expression patterns of many genes are changed (*Mok et al., 2015*). *Pb*RC private variants were identified in genes mutated in gametocyte non-producing lines derived from the reference strain *Pb*ANKA (*Sinha et al., 2014*), genes with annotated function with respect to blood-stage growth (*Bushell et al., 2017*) and *P. berghei* orthologs of genes with altered expression in ART-resistant *P. falciparum* (*Mok et al., 2015*). The summaries of *Pb*RC private variants for each phenotypic category are shown in Table 3, and all variants in each phenotypic category are shown in Table S3. Non-synonymous variants are more prevalent than synonymous for each phenotypic category, including essential genes.

**Table 3 Summaries of *P. berghei* RC private variants located in genes associated with different phenotypes.**

| | Missense variants | Synonymous variants | Modifier variants[a] | Stop variants[b] | No. genes with variants |
|---|---|---|---|---|---|
| All genes | 2,019 | 1,458 | 55 | 6 | 1,739 |
| Core[c] | 1,980 | 1,438 | 53 | 5 | 1,705 |
| Gametocytogenesis[d] | 90 | 55 | 1 | 0 | 35 |
| Essential[e] | 563 | 435 | 8 | 2 | 484 |
| Slow[e] | 228 | 153 | 3 | 2 | 166 |
| Dispensable[e] | 380 | 280 | 11 | 4 | 342 |
| Fast[e] | 3 | 2 | 0 | 0 | 1 |
| ART-R[f] | 415 | 289 | 16 | 3 | 344 |

Notes:

A total of 3,538 single nucleotide variants (SNV) private to the *P. berghei* RC strain after filtering and located within annotated genes were identified by whole-genome sequencing comparing with the reference strain *Pb*ANKA v3 and other strains reported in *Otto et al. (2014)*. The numbers of SNV in each functional category annotated by the SnpEff software version 3.6g are indicated in the columns. The numbers of variants in genes associated with different phenotypes are shown in each row. Note that some genes are associated with more than one phenotype.

[a] Includes the following SnpEff functional categories: splice region variant and intron variant, non-coding transcript exon variant, splice region variant and stop retained variant, splice region variant and synonymous variant and modifier synonymous stop variant.

[b] Includes all stop codon variants predicted to alter the open reading frame.

[c] *P. berghei* genes orthologous across rodent and primate malaria *Plasmodium* spp. (*Otto et al., 2014*).

[d] Genes mutated in gametocyte non-producing lines of *P. berghei* (*Sinha et al., 2014*).

[e] Growth phenotypes annotated from *P. berghei* gene knockout mutants in the PlasmoGEM database (*Bushell et al., 2017*).

[f] *P. berghei* orthologs of *P. falciparum* genes with altered gene expression profiles in artemisinin-resistant isolates (*Mok et al., 2015*).

Finally, drug resistance in *Plasmodium* spp. is often modulated by CNVs; for instance, the *P. falciparum* gene *pfmdr1* is amplified in geographical regions where antimalarial resistance is common (*Nair et al., 2006*). CNVs among *P. berghei* strains were identified from whole-genome sequencing data (Table S4). All of the CNVs are present in chromosomal regions near chromosome ends. All of the genes that overlap CNVs are members of multigene families. Amplified regions with copy number greater than one are not private to any strain, including *Pb*RC. Large deletions (>1 kb) are prevalent in all strains except *Pb*SP11_A, and *Pb*RC has the greatest number of large deletions.

## DISCUSSION

Our analysis of *P. berghei* genomes revealed that the majority of SNVs are private to each strain, in which *Pb*RC is most diverged strain of all. Moreover, the genome-wide pattern of negative Tajima's D score (Fig. 3) can be interpreted as evidence of an excess of rare variants because of a recent population bottleneck. Other *Plasmodium* spp. also show overall negative Tajima's D score (*Parobek et al., 2016*). The overall negative Tajima's D score (and other measures (*Rutledge et al., 2017*; *Otto et al., 2016*)), are consistent with recent population bottlenecks in *Plasmodium* spp. Although Tajima's D score is negative overall for *P. berghei*, we identified some genes with high Tajima's D scores. Some of these genes are known to function in host cell invasion, including *Pb*CTRP, berghepain 1, *Pb*GAP40 and *Pb*GAC. The *P. berghei* CTRP protein is important for mosquito midgut invasion by the ookinete (*Dessens et al., 1999*). The berghepain 1 gene may also function in

midgut invasion, as mutants of the *P. falciparum* homologous gene (falcipain 1) are defective for oocyst production (*Eksi et al., 2004*). The GAP40 and GAC genes may function as part of the glideosome complex, an actin- and myosin-based machine conserved across Apicomplexa that powers parasite motility, migration, host cell invasion and egress (*Frénal et al., 2010*; *Jacot et al., 2016*). The high Tajima's D scores for these genes suggests balancing selection could operate in *P. berghei* to favor alleles that produce antigenically diverse proteins allowing invasive parasites to evade host immune systems.

The large number of variants private to *Pb*RC makes it difficult to pinpoint causal variants of its drug-resistant phenotype. Moreover, the N strain progenitor from which *Pb*RC was derived by CQ selection was not available to us, and forward genetic mapping is not possible since *Pb*RC does not produce gametocytes (*Peters, Fletcher & Staeubli, 1965*). Therefore, we manually curated variants private to *Pb*RC to identify those in candidate genes known to modulate drug sensitivity in other *Plasmodium* spp. The level of GSH modulates CQ sensitivity in *P. falciparum*, which is controlled by the level of expression of $\gamma$-glutamyl cysteine synthetase enzyme (*Ginsburg & Golenser, 2003*). *Pb*RC carries a I413K variant in the *Pbγ-gcs* gene encoding this enzyme (Table 2). However, *P. berghei* parasites with knockout of *Pbγ-gcs* do not show altered sensitivity to CQ or ART (*Vega-Rodríguez et al., 2015*; *Songsungthong et al., 2016*). Furthermore, the *Pbγ-gcs* I413K variant may have pre-existed in the CQ-sensitive N strain progenitor (*Pérez-Rosado et al., 2002*), and so is not likely to modulate drug sensitivity.

*Pb*RC harbors five non-synonymous variants in the *Pbubp1* gene. The homologous gene is mutated in drug-resistant *P. chabaudi* (*Hunt et al., 2007*; *Henriques et al., 2013*). The *ubp1* mutated residues in drug-resistant *P. chabaudi* are located in a conserved putative ubiquitin binding region of the protein, which may disrupt its function and lead to increased proteasomal degradation of UBP-1 substrates (*Hunt et al., 2007*). However, none of the *Pbubp1* variants in *Pb*RC correspond to the *ubp1* mutations in *P. chabaudi*, and except for N2279D, the *Pbubp1* variant residues are not conserved among *Plasmodium* spp. Therefore, it is not known if the *Pbubp1* variants in *Pb*RC affect the function of the protein. Disruption of ubiquitination pathways has been implicated in CQ and ART resistance mechanisms, although such disruption could occur by mutations in different genes.

Genome sequencing of the CQ- and ART-resistant clone AS-ART isolated in *Afonso et al. (2006)* revealed a non-synonymous mutation in the AP2-μ gene, which was not present in the AS-15CQ progenitor strain (*Henriques et al., 2013*). Cross resistance to CQ and ART was thus attributed to the presence of *ubp1* and AP2-μ mutations in AS-ART (*Henriques et al., 2013*). AP2-μ gene mutation in the AS-ART parasite was proposed to change the balance of endocytosis toward a clathrin-independent pathway (*Henriques et al., 2013*). In this scenario, endocytosis of hemoglobin may be reduced with subsequent lower production of heme catabolite and reduced CQ and ART efficacy. However, no variants in the *P. berghei* AP2-μ homologue were found in *Pb*RC, suggesting that resistance pathways differ between AS-ART and *Pb*RC. Mutations in the *P. falciparum* K13 gene are thought to lead to important changes in ubiquitination patterns that affect sensitivity to ART (*Dogovski et al., 2015*); however, no *Pb*RC variants were found in the

homologous gene *Pb*K13. The activity of *Pf*PI3K modulates ART sensitivity in *P. falciparum*, which is controlled by K13-mediated ubiquitination (*Mbengue et al., 2015*). *Pb*RC harbors a F320C variant in the homologous *Pbpi3k* gene. The F320C variant is located in an *N*-terminal domain not present in homologous PI3K proteins with reported structures (i.e., human and *Drosophila*); hence, it is difficult to predict if this mutation affects *Pb*PI3K protein function.

Mutations in the *Pfcrt* gene modulate CQ sensitivity in *P. falciparum*, and a V42F variant was found in the homologous gene (*Pbcrt*) of *Pb*RC (Table 2). The *Pbcrt* residue 42 is predicted to be in the *N*-terminal cytosolic part of the protein before the first transmembrane domain; however, the equivalent residue in *Pfcrt* is not reported as mutated among field isolates of CQ-resistant *P. falciparum* (*Ecker et al., 2012*). It should be noted that other *crt* mutations, such as the C101F variant in piperaquine-resistant *P. falciparum* parasites (*Dhingra et al., 2017*) can cause defects in food vacuole morphology similar to *Pb*RC. It is difficult to predict the effect of the V42F variant on *Pbcrt* function by comparison with *Pfcrt*, since allelic replacement of *Pbcrt* with *Pfcrt* from CQ-resistant *P. falciparum* modulated CQ sensitivity only in sexual stages of *P. berghei*, pointing to divergence of *crt* gene function between the two species (*Ecker et al., 2011*). The degree of CQ sensitivity in *P. falciparum* with *Pfcrt* mutation is modulated by variants in other genes such as *Pfmdr1*, as shown recently by genome editing studies (*Veiga et al., 2016*). The *Pb*RC *Pbmdr1* V54A variant residue is not equivalent to any *Pfmdr1* residue shown to modulate CQ or ART sensitivity (*Veiga et al., 2016*); hence, it is difficult to predict the effect of the V54A variant on *Pbmdr1* function.

The *Pb*RC strain produces less hemozoin than other CQ-sensitive strains (*Peters, 1964*; *Peters, Fletcher & Staeubli, 1965*), which may be due to defective hemoglobin digestive enzymes. A protein complex of hemoglobin digestive enzymes has been described in *P. falciparum* (*Chugh et al., 2013*). This complex may have a simpler composition in *P. berghei*, since this species possesses only one digestive falcipain-like enzyme (berghepain-2), one plasmepsin (plasmepsin IV) and one hemoglobin digestive protein (HDP) (*Ponsuwanna et al., 2016*). No variants were found among these genes in *Pb*RC, and of the downstream digestive enzymes, only one variant in the *Pbdpap3* gene was found. This gene is non-essential in *P. berghei* (*Lin et al., 2015*), and so mutation of this gene may play a minor role in modulating CQ and ART sensitivity.

We extended the candidate gene approach toward association of *Pb*RC private variants with other phenotypic traits (Table 3; Table S3). The inability of the *Pb*RC strain to produce gametocytes is likely due to non-synonymous mutations in genes previously shown to be mutated in gametocyte non-producing lines (*Sinha et al., 2014*). Most importantly, among these genes the transcription factor AP2-G known to be essential for gametocytogenesis harbors the missense mutations P86S, S159N and F546V. Using the same reasoning, the slow growth of *Pb*RC is likely caused by the many non-synonymous mutations in genes which are known to cause growth defects when knocked out (*Bushell et al., 2017*). The variants in genes causing growth defects may also contribute to the drug-resistant trait, since nearly a third of genes orthologous with those with altered

expression in ART-resistant *P. falciparum* isolates (*Mok et al., 2015*) are also essential (*Bushell et al., 2017*).

In addition to non-synonymous SNVs, CNVs could contribute toward some of the *Pb*RC phenotypic traits. However, no CNVs were found in core genes that could modulate antimalarial sensitivity or growth. CNVs were found only among sub-telomeric genomic regions encompassing multigene families, which were reported previously to be variable among *P. berghei* strains (*Otto et al., 2014*). Large deletions appear to be more prevalent in sub-telomeric regions in *Pb*RC compared with other strains, although we are cautious in making any inferences from these patterns owing to potential sequence alignment error in these genomic regions.

## CONCLUSION

Analysis of genomic variants across six strains of *P. berghei* revealed an excess of rare variants, consistent with a population bottleneck as reported for other *Plasmodium* spp. Several variants were identified as private to *Pb*RC which could modulate drug sensitivity, although direct testing of these variants using approaches such as genome editing is necessary to test causality.

## ACKNOWLEDGEMENTS

The following reagents were obtained through BEI Resources, NIAID, NIH: *Plasmodium berghei*, Strain (ANKA) 507m6cl1, MRA-867, contributed by C.J. Janse and A.P. Waters and *Plasmodium berghei* RC, MRA-404, deposited by W. Peters and B.L. Robinson. We thank Dafra Pharma for artesunate, Dr. Thomas Otto for suggestions on genome analysis, Miss Wanrisa Khamtawee for technical assistance, the reviewers for their suggestions, and the curators of PlasmoDB and GeneDB.

### Funding

This work was supported by the National Center for Genetic Engineering and Biotechnology (BIOTEC), Thailand, (grant nos. P1300635 and P1300105 to WS), BIOTEC Platform Technology, Thailand (no. P1551103 to PJS), the Howard Hughes Medical Institute, USA (no. 55005512 to SK), the National Science and Technology Development Agency's Cluster Program Management, Thailand (no. P1450883 to SK) and the Thailand Research Fund (no. RSA5880064 to CU and no. RSA5860081 to ST). The funders had no role in study design, data collection and analysis, decision to publish, or preparation of the manuscript.

### Grant Disclosures

The following grant information was disclosed by the authors:
National Center for Genetic Engineering and Biotechnology (BIOTEC), Thailand: P1300635 and P1300105.
BIOTEC Platform Technology, Thailand: P1551103.

Howard Hughes Medical Institute, USA: 55005512.
National Science and Technology Development Agency's Cluster Program Management, Thailand: P1450883.
Thailand Research Fund: RSA5880064 and RSA5860081.

## Competing Interests

The authors declare that they have no competing interests.

## Author Contributions

- Warangkhana Songsungthong conceived and designed the experiments, performed the experiments, analyzed the data, wrote the paper, prepared figures and/or tables, reviewed drafts of the paper.
- Supasak Kulawonganunchai analyzed the data, prepared figures and/or tables.
- Alisa Wilantho analyzed the data, prepared figures and/or tables.
- Sissades Tongsima analyzed the data.
- Pongpisid Koonyosying performed the experiments.
- Chairat Uthaipibull contributed reagents/materials/analysis tools, reviewed drafts of the paper, project administration and funding acquisition.
- Sumalee Kamchonwongpaisan contributed reagents/materials/analysis tools, reviewed drafts of the paper, project administration and funding acquisition.
- Philip J. Shaw conceived and designed the experiments, analyzed the data, wrote the paper, prepared figures and/or tables, reviewed drafts of the paper.

## Animal Ethics

The following information was supplied relating to ethical approvals (i.e., approving body and any reference numbers):

This study was carried out in accordance with the guidelines in the Guide for the Care and Use of Laboratory Animals of the National Research Council, Thailand. All animal experiments were performed with the approval of BIOTEC's Institutional Animal Care and Use Committee (Permit number BT-Animal 02/2557).

## Data Availability

The raw data are deposited in the NCBI Sequence Read Archive, accession number PRJNA277169. All details of publicly available bioinformatic tools used to analyse the data are provided in the manuscript.

## Supplemental Information

Supplemental information for this article can be found online at http://dx.doi.org/10.7717/peerj.3766#supplemental-information.

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
