# Peer review of "The Plasmodium berghei RC strain is highly diverged and harbors putatively novel drug resistance variants"

_PeerJ, doi:10.7717/peerj.3766_

## Round 0.1 · original submission · Minor Revisions

Overall, the reviewers were quite positive. I believe that the queries that were raised should be addressable in a straightforward and timely manner, and we look forward to seeing your revised manuscript.

·

Basic reporting

No additional comment; the paper satisfies all the criteria.

Experimental design

No comment.

Validity of the findings

No comment.

Additional comments

In this study, Songsungthong et al use whole genome sequencing to examine the P. berghei RC (PbRC) strain that in earlier work by Peters was selected to be chloroquine resistant. The RC strain is deficient in haemozoin formation, has increased levels of glutathione, and was more recently identified as resistant to artemisinin. Here, the authors start by validating the reduced susceptibility to artesunate, with the RC strain approximately 8-fold less sensitive than PbANKA. Whole genome sequencing of PbRC revealed a high level of divergence from the reference PbANKA, as well as several other sequenced P. berghei genomes. The high number of variants that are unique to PbRC, almost 10-fold higher than any other strain with unique or private variants, confounds the identification of possible resistance loci that could contribute to the observed phenotypes. The authors survey possible candidates, and report SNVs in several, including PbCRT, PbMDR1, the deubiquitinating enzyme PbUBP1, and PbPI3K, but not the P. berghei ortholog of Kelch13. None of these SNVs is an obvious smoking gun, and the inability of the RC strain to form gametocytes precludes the use of genetic crosses to further dissect the resistance phenotypes.

Overall the study by Songsungthong et al demonstrate the high level of divergence of the RC strain from other sequenced strains of P. berghei, and provide a foundation from which further genetic validation of drug resistance candidates could be based.

There are a few additional points that the authors should consider. i) For the analysis of “private” variants in PbRC, it would be informative if the authors described in more detail the proportion of SNVs that are located in coding regions, are non-synonymous, and are in the core genome. ii) Gene amplification is a common mechanism of drug resistance in Plasmodium sp. Could the authors comment on whether gene amplifications could be detected, and might be a mechanism of resistance in this strain? iii) Given the inability of the RC strain to form gametocytes, were there mutations detected in AP2-G and other candidate loci involved in gametocytogenesis?

·

Basic reporting

Well written paper that is clear and easy to understand. All the data has been shared.

Experimental design

Antimalarial drug resistance is an emerging global health issue as malaria has become untreatable in some parts of the world due to failure of drug treatment, including the current frontline therapy, artemisinin. Recent evidence show that artemisinin resistance is multi-factorial though mutations in one gene, kelch 13, show a clear association with resistance. The authors of this study attempt to identify the genetic basis of antimalarial resistance using the murine malaria parasite, Plasmodium berghei. A specific strain of P. berghei, PbRC was known to be resistant to several antimalarial drugs, including artemisinin (ART) and chloroquine (CQ). The authors hypothesize that PbRC may share some of the resistance mechanisms with drug resistant human malaria parasites. The authors use whole genome sequencing to answer this question and compare the genome of PbRC to published genomes of several P. berghei strains to identify SNPs unique to PbRC. They use Tajima's D score to identify genes undergoing selection. The methods are described in detail and the work is well done. The story of artemisinin resistant malaria is still evolving and we don't have the complete picture. The authors identify numerous genes that may play a role in drug resistance.

Validity of the findings

The authors show that PbRC is resistant to artesunate, a water-soluble derivative of ART, with an increased ED50. Using whole genome sequencing, the authors identify several unique SNPs in the PbRC strain and convincingly demonstrate that it is quite divergent from other sequenced P. berghei strains. Unfortunately, the SNPs identified in PbRC do not match with SNPs found in artemisinin resistant P. falciparum. This does not take away from the study though as artemisinin resistance is known to be multi-factorial and our understanding of it is incomplete. The authors are careful about using the literature to make an educated guess as to which sets of genes are likely responsible for drug resistance.
However, one issue is that the PbRC strain does not form gametocytes and strains that do not form gametocytes often have associated mutations with them. The authors do not compare PbRC genome with the genome of the ANKA 2.33 strain (gametocyte non-producer) to narrow down their targets.
Another potentially useful method to narrow down the list of candidate genes responsible for drug resistance would be to assess essentiality. The authors do that for one candidate (Pbdpap3) but not for others. As the authors may be aware, this information is available for nearly 3000 genes in the P. berghei genome (Plasmogem phenotype database: http://plasmogem.sanger.ac.uk/phenotypes).
These comparisons may help the discussion to focus on drug resistance and curate the list of candidates further.

Additional comments

Please discuss why you think that host cell invasion genes are changing due to balancing selection. Its unclear how they would help/hurt drug resistance. Why would PbRC need to evade the immune system more than other strains? Maybe it has more to do with being a gametocyte non-producer than drug resistance?
It may also be useful to compare the unique PbRC SNPs with the transcriptional dataset for ART-resistant P. falciparum (Mok et al Science 2015, 347:431-35).

---

## Round 0.2 · accepted · Accept

Nice job handling the reviewer queries. Congrats on your work.